# Effect of Different Herbage Allowances from Mid to Late Gestation on Nellore Cow Performance and Female Offspring Growth until Weaning

**DOI:** 10.3390/ani14010163

**Published:** 2024-01-04

**Authors:** Luciana Melo Sousa, William Luiz de Souza, Karla Alves Oliveira, Iorrano Andrade Cidrini, Philipe Moriel, Henrique César Rodrigues Nogueira, Igor Machado Ferreira, Germán Dario Ramirez-Zamudio, Ivanna Moraes de Oliveira, Laura Franco Prados, Flávio Dutra de Resende, Gustavo Rezende Siqueira

**Affiliations:** 1Departament of Animal Science, São Paulo State University, Jaboticabal 14884-900, SP, Brazil; williamluizdesouzaa@gmail.com (W.L.d.S.); karla.oliveira@unesp.br (K.A.O.); iorranoandrade@gmail.com (I.A.C.); igor.machado@unesp.br (I.M.F.); flaviodutraderesende@gmail.com (F.D.d.R.); siqueiragr@gmail.com (G.R.S.); 2Agência Paulista de Tecnologia dos Agronegócios, Colina 14770-000, SP, Brazil; henriquecrn.contato@gmail.com (H.C.R.N.); imoraesdeoliveira@yahoo.com.br (I.M.d.O.); laurafrancoprados@hotmail.com (L.F.P.); 3Range Cattle Research and Education Center, University of Florida, Ona, FL 33865, USA; pmoriel@ufl.edu; 4Departament of Animal Science and Food Engineering, São Paulo University, Pirassununga 13635-900, SP, Brazil; germanramvz@gmail.com

**Keywords:** female calves, fetal programming, forage allowance, milk yield, Nellore

## Abstract

**Simple Summary:**

Many studies describe the importance of granting sufficient nutrients to pregnant cows to modulate offspring growth. However, studies evaluating the capacity of tropical forages to supply all the nutrients needed for fetal development with Nellore cows are scarce. These studies are essential to understand how different herbage conditions affect the performance of cows during the gestational period. This study evaluated if the changes in herbage allowance can modulate fetal development and its effects on the cow–calf pair until weaning. Our results showed that reducing the maternal herbage allowance decreased cow prepartum performance, postpartum milk yield, and milk composition, which modulate offspring preweaning growth.

**Abstract:**

This study evaluated different herbage allowances from mid to late pregnancy on pre- and postpartum physiological responses, milk production, and the performance of Nellore cows and the preweaning growth of their female offspring. Sixty multiparous Nellore cows were blocked by their body weight (BW; 425 ± 36 kg) and body condition score (BCS; 3.67 ± 0.23, scale 1–5) and randomly allocated to twelve pastures. Treatments consisted of two different herbage allowances (HA) during pregnancy: low HA (LHA; 2.80 kg DM/kg of BW) and high HA (HHA; 7.60 kg DM/kg of BW). Both treatment groups were fed 1 g/kg BW of a protein supplement. After calving, all cow–calf pairs were combined in a single group. The effects of maternal treatment × day of the study were detected for herbage mass and allowance, the stocking rate and forage crude protein, and for cow BW, BCS, and carcass measures (*p* < 0.01). Milk yield corrected to 4% fat, while the levels of fat total solids and cow plasma IGF-1 and urea were different (*p* ≤ 0.04) between treatments. HHA offspring was heavier (*p* ≤ 0.05) at 120 days and at weaning. A high herbage allowance can be implemented from mid-gestation until calving to increase cow prepartum performance, post-partum milk yield and composition, and positively modulate female offspring preweaning growth.

## 1. Introduction

Seasonality in annual forage production leads to oscillations in herbage mass and forage nutrient composition, with 70 to 80% of herbage mass production occurring during the rainy season compared to 20 to 30% during the dry season [1]. Therefore, grazing, spring-calving beef cows are often subjected to nutrient restriction during the second and third trimesters of gestation [2]. Studies evaluating the impacts of maternal nutrition during pregnancy on progeny development and performance have been carried out mostly with *Bos taurus* breeds [3] and very few studies with Nellore (*Bos indicus*) cows in tropical/subtropical environments [4,5,6]. Additional studies using a wider variety of nutritional strategies during pregnancy (such as forage amount and type) of Nellore cows are crucial due to the wide differences in metabolism, behavior, and performance between *Bos taurus* and *Bos indicus* breeds following similar management [7,8].

Diets based only on forage usually do not supply all the nutrients required by grazing pregnant beef cows. This situation can be changed by supplementing deficient pasture nutrients such as proteins and minerals [9]. It was observed that when Nellore cows receive an ad libitum amount of herbage mass (>4700 kg DM/kg BW) and consequently adequate herbage allowance (HA) combined with protein supplementation during late pregnancy, they might not experience severe nutrient restriction during pregnancy [6].

Herbage allowance can drive forage dry matter (DM) intake and productive performance [10], and combined with forage nutritive value, might lead to nutrient surplus or restriction during pregnancy, impacting progeny in utero development and postnatal performance. It was hypothesized that calves born from cows assigned to low HA (in utero nutrient restriction) could exhibit poorer preweaning growth performance compared to cohorts born from cows assigned to a high HA (in utero nutrient surplus). Therefore, our objectives were to evaluate the effects of contrasting HA (low vs. high) from mid-gestation to calving on the performance, physiology, and milk production of Nellore cows and the postnatal growth and physiology of their offspring.

## 2. Materials and Methods

The current study was conducted at the Experimental Farm of Agência Paulista de Tecnologia dos Agronegócios, Regional Center of Colina (APTA-Colina), Colina, SP, Brazil (20°43′5″ S and 48°32′38″ W) from June 2019 to July 2020. The procedures used in this study were approved by the Ethics Committee on Animal Use of Development Decentralization Department (CEUA-DDD, protocol no 0004/2020).

### 2.1. Animals and Diets

#### Prepartum (Day 0 to 150) and Preweaning (Day 150 to 390)

On day 0 of the study (approximately 150 ± 11 d before calving), 60 multiparous (4.5 ± 1.5 years of age) Nellore cows that were pregnant with female calves via artificial insemination (from 3 bulls) were stratified by their initial body weight (BW = 425 ± 36 kg) and body condition score (BCS = 3.67 ± 0.23) and then assigned to 12 pastures (5 cows/pasture) in a randomized block design with 6 blocks, where each pasture represented an experimental unit. The pastures were composed of *Urochloa brizantha* cv. Marandu (7.5 hectare/pasture). Maternal treatments were randomly assigned to the pastures within each block (6 pastures/maternal treatment) and consisted of two HA: low (LHA; 2.80 kg of forage DM/kg BW) and high (HHA; 7.60 kg of forage DM/kg BW; [11]). Additional pregnant cows were used to alter the stocking rates (SR; 3.2 and 1.6 animal units/ha for LHA and HHA; 1 animal unit = 450 kg) to achieve the target HA during the prepartum period. A continuous stocking method was applied throughout the study, with a fixed number of cows per pasture (24 and 12 cows for LHA and HHA). All cows received a daily protein supplementation at 0.10% of BW from day 0 to 150. Supplements were offered at 10 am in feeders located at the pasture. The cows were checked once daily for calving and calved on average on day 150 ± 11 of the study. The first offspring (calves in utero when maternal treatments were provided) were weighed within the first 12 h of life. From day 150 to 390, all cow–calf pairs were combined in a single group and transferred to a single pasture (area: 10 ha; HA: 2.36 ± 0.39 kg DM/kg BW; SR: 7.14 ± 1.38 AU/ha; herbage mass: 7332 ± 1081 kg DM/ha; CP: 6.57 ± 1.94% of DM) and managed similarly with free choice access to trace mineral salt supplementation until weaning (Table 1). All calves were weaned on day 390. Calf health was monitored daily by trained personnel from birth until the end of the study. Due to unexpected sexing errors, only 42 cow–calf pairs (3 to 5 cows/pasture) were used in the statistical evaluation.

### 2.2. Data Collection

#### 2.2.1. Forage and Feed

The herbage mass and hand-plucked samples of pastures were collected every 35 d from day 0 to 150 and then every 42 d from day 150 to 390 [12,13]. The mass was analyzed using the double-sampling method [11]. To determine the forage mass of each pasture, it was associated with forage height readings (100 points of normal and compressed height—using the rising plate meter). In each pasture, 3 points of low height were selected, including medium and high (determined according to ±2 standard errors). Afterwards, linear regression equations were carried out using the mass data and their respective heights, where a relationship was established between the forage height and the forage mass. Herbage allowance was determined every 35 d from day 0 to 150 [11]. Samples of supplements were collected every 35 d from day 0 to 150. Immediately after collection, all forage and supplement samples were dried at 55 °C for 72 h using a forced-air oven and then ground in a Wiley mill (Thomas Model 4, Thomas Scientific, Swedesboro, NJ, USA) to pass through a 1 mm mesh sieve before further analysis.

#### 2.2.2. Maternal

The individual shrunk BW of cows was measured on days 0 (start of treatment supplementation), 130 (near calving), and 390 (weaning) with a digital scale, followed by feed and water restriction for 16 h. Cow’s individual BCS (scale 1–5; [14]) were assessed on days 0, 35, 70, 130, 150, 203 and 390. Calving dates were recorded for each cow. Blood samples (10 mL) were collected from the jugular vein from 18 cows per treatment (3 cows/pasture; randomly selected on day 0) into commercial tubes (BD Vacutainer^®^ SST II Advance) containing 158 USP of sodium-heparin on days 0 and 130 to determine the plasma concentrations of glucose, urea, albumin, creatinine, total proteins, cholesterol, triglycerides, amino aspartate-transferase (AAT), gamma-glutamyl transferase (GGT), insulin and insulin-like growth factors 1 (IGF-1). All blood samples were collected before morning supplementation and were immediately placed on ice following collection and then centrifuged at 3000× *g* for 15 min at 4 °C. Plasma samples were stored frozen at −20 °C until laboratory analysis. Cows selected for blood sampling were randomly chosen in a manner that represented the average cow BCS, BW, age, and calving date of each respective treatment.

A carcass ultrasound of all cows was performed on days 0, 130, and 390 using a veterinary ultrasound Piemedical—Scanner 200—with a linear probe (ASP-18) and frequency of 3.5 MHz. The longissimus muscle area (LMA; cm^2^), backfat thickness (BFT, mm), and marbling (points 1 to 10) were measured through ultrasound images. Images were taken between the 12th and 13th ribs, transverse to the *Longissimus dorsi* muscle. Thus, the fat thickness was measured in the distal middle third of the rib eye area. Rump fat thickness (RFT) was measured in the longitudinal position between the ileum and ischium bones in the rump (the junction of the gluteus medius and biceps femoris muscles). Vegetable oil was used as an acoustic coupling in all evaluations.

Cow milk production was collected on days 180, 270, and 360 from 3 cows/pasture (the same cows assigned to blood collection). The cows were separated from their calves at 05 p.m. At 06 a.m. of the following day, the cows were injected with 2 mL of oxytocin (10 IU/mL; Ocitovet^®^, Jacareí, Brazil) in the mammary artery and immediately milked using a milking machine. The exact time when each cow was milked was recorded, and the daily yield (DY) was calculated according to [15]: DY (kg/day) = (MoY × 24 h)/MMT, where MoY represents the morning yield (kg), MMT represents the morning milking time (kg), and where MMT = (morning milking time recorded—the time recorded of previous calf separation). The milk yield was corrected to 4% of fat-corrected milk (FCM) according to [16]: FCM (kg/day) = 0.4 × DY (kg/day) + 15 × milk fat content (kg/day). Individual milk samples (50 mL) were collected in bottles containing a preservative (bromonate) to evaluate the milk’s composition. Milk samples were stored under refrigeration immediately following collection and then sent to a commercial laboratory (Milk Clinic, Piracicaba—Brazil) and analyzed for concentrations of protein, fat, lactose, total solids, ureic N and casein contents, and somatic cell content (SCC).

#### 2.2.3. First Offspring

The individual unshrunk BW of female calves were collected within 12 h after birth and on days 270 and 390. The body weight at weaning (day 390) was also adjusted to 205 days of age. Blood samples (10 mL) were collected from the jugular vein from 18 heifer calves per treatment (3 heifer/pasture) into commercial tubes (BD Vacutainer^®^ SST II Advance, Curitiba, Brazil) containing 158 USP of sodium-heparin on days 270 and 390 to determine the plasma concentrations of glucose, urea, albumin, creatinine, total proteins, cholesterol, triglycerides, amino aspartate-transferase (AAT), gamma-glutamyl transferase (GGT), insulin and insulin-like growth factors 1 (IGF-1). All blood samples were collected in the morning and were immediately placed on ice following collection and then centrifuged at 3000× *g* for 15 min at 4 °C. Plasma samples were stored frozen at −20 °C until laboratory analysis.

A carcass ultrasound of all heifers was performed on day 390 using a veterinary ultrasound Piemedical—Scanner 200—with a linear probe (ASP-18) and frequency of 3.5 MHz. The longissimus muscle area (LMA; cm^2^), backfat thickness (BFT, mm), and marbling (points 1 to 10) were measured through ultrasound images. Images were taken between the 12th and 13th ribs, transverse to the *Longissimus dorsi* muscle. Thus, the fat thickness was measured in the distal middle third of the rib eye area. Rump fat thickness (RFT) was measured in the longitudinal position between the ileum and ischium bones in the rump (the junction of the gluteus medius and biceps femoris muscles). Vegetable oil was used as an acoustic coupling in all evaluations.

#### 2.2.4. Laboratory Analysis

Forage and supplement samples were dried at 55 °C for 72 h in a forced-air oven and ground in a Wiley mill (Thomas Model 4, Thomas Scientific, Swedesboro, NJ, USA) using a 1 mm screen. Samples (1 mm processed) were analyzed in duplicates at the laboratory of the research unit for CP (N ×6.25; Kjeldahl method 984.13; [17]).

The plasma samples of cows and calves were analyzed in duplicate to assess the concentrations of glucose (K-082–3; Bioclin), urea (K-056; Bioclin), albumin (K-040; Bioclin), creatinine (K-222; Bioclin), total proteins (K-031; Bioclin), cholesterol (K-083; Bioclin), triglycerides (K-117; Bioclin), amino aspartate-transferase (K-048–6; Bioclin), and gamma-glutamyl transferase (K-080–2; Bioclin) using commercial kits with absorbances measured on a spectrophotometer (SBA 200, Celm). In addition, IMMULITE^®^ 1000 commercial IGF-1 and insulin kits were used, where the readings were performed by chemiluminescence obtained via the Dimension EXL 200 Integrated Biochemistry system (Siemens Healthcare Diagnostics, Munich, Germany). All blood assays were performed at the Animal Biochemistry and Physiology Laboratory (LBFA) at the São Paulo University—Department of Nutrition and Animal Production—SP, Brazil.

Milk samples were analyzed for protein, fat, lactose, total solids, ureic nitrogen, and casein contents using infrared spectroscopy (MilkoScan FT1, ISO 9622/IDF 141 anchored by calibration to reference methods [18]), and the somatic cells content (SCC) was obtained via the flow cytometric method Bentley Combo System 2300 (method ISO 13366-2/IDF 148-2 anchored by calibration to reference methods [19]) at Milk Clinic at the São Paulo University—Piracicaba—SP, Brazil.

### 2.3. Statistical Analyses

Binary data (the cow pregnancy rate) were analyzed using the GLIMMIX procedure of SAS (SAS Institute Inc., Cary, NC, USA, version 9.4). Nonbinary data were analyzed as a complete block design using the MIXED procedure of SAS (version 9.4). The pasture was considered the experimental unit, whereas the cow (pasture) or calf (pasture) were included as random effects in all statistical analyses. The cow BW, BCS, ultrasound carcass, offspring BW and blood data, milk yield, and herbage characteristics were analyzed as repeated measures and tested for the fixed effects of maternal treatments, the day of the study, and all resulting interactions, using the cow (pasture) or calf (pasture) as the subjects, except for herbage characteristics which included pasture as the subject. The covariance structure was chosen using the lowest Bayesian information criterion. The cow and offspring average daily gain (ADG), cow blood data, and offspring ultrasound carcass were tested for the fixed effects of maternal treatment using the cow (pasture) or calf (pasture) as a random effect. Other potential effects were evaluated in the model, such as the initial measure of each variable as a covariate (*p* ≤ 0.05). This criterion was used due to sexing errors that led to a change in the number of experimental animals and, consequently, to the variation in the initial measurements. For the variables measured after calving, the initial BW of the cow was used as a covariate (*p* ≤ 0.05). All results are reported as the least-square means. The means were separated by PDIFF when a significant F-test was detected. Significance was fixed at *p* ≤ 0.05 and tendencies when *p* > 0.05 and ≤0.10.

## 3. Results

### 3.1. Pasture

The effects of maternal treatment × the day of the study were detected (*p* ≤ 0.05) for the herbage mass, HA, stocking rate, and crude protein (Figure 1a–d). Herbage mass on days 35, 105 and 140 were lower (*p* ≤ 0.01) for LHA vs. HHA. On day 70, herbage mass tended to be lower (*p* = 0.08) for LHA vs. HHA (Figure 1a). The herbage allowance and stocking rate were different between maternal treatments (*p* < 0.01) from d 0 to 140. Herbage allowance was lower for LHA vs. HHA (Figure 1b), while the stocking rate was higher for LHA vs. HHA (Figure 1c). Forage CP on days 35, 70, 105, and 140 was lower (*p* ≤ 0.04) for LHA vs. HHA (Figure 1d). Herbage mass and forage CP on day 0 did not differ (*p* > 0.70) between maternal treatments.

### 3.2. Prepartum (Days 0 to 150)

The effects of maternal treatment × the day of the study were observed (*p* < 0.01) for the cow’s BCS and BW (Table 2). Cow BW on days 130 and 390 was less (*p* < 0.01) for LHA vs. HHA cows. Cow BCS on days 130 and 150 was greater (*p* ≤ 0.01) for HHA vs. LHA cows. Cow BCS on days 0, 70, 203, and 390 did not differ (*p* ≥ 0.14) between maternal treatments, whereas cow BCS on day 35 tended to be greater (*p* = 0.07) for HHA vs. LHA cows. The effect of maternal treatment was detected (*p* ≤ 0.01) for cows’ ADG from day 0 to 150 and day 150 to 390 (Table 2). Cow ADG from day 0 to 150 was greater (*p* < 0.01) for HHA vs. LHA cows, whereas from day 150 to 390, it was lower (*p* = 0.01) for HHA vs. LHA cows. The effects of maternal treatment × the day of the study were detected (*p* ≤ 0.02) for LMA, BFT, and RFT (Table 2). Cow LMA, BFT, and RFT on day 130 were lower (*p* ≤ 0.01) for LHA cows but did not differ on days 0 and 390 (*p* ≥ 0.54) between maternal treatments. The cow marbling and pregnancy rate in the subsequent breed season did not differ (*p* ≥ 0.15) between maternal treatments (Table 2).

The effects of maternal treatment × the day of the study were not detected (*p* ≥ 0.34) for milk yield, fat-corrected milk yield, and milk contents, but the effects of maternal treatment were detected (*p* ≤ 0.03) for fat-corrected milk yield, milk fat and total solids (Table 3). Fat-corrected milk, milk fat, and total solids were greater for HHA vs. LHA cows. The effect of the day of study was detected (*p* ≤ 0.02) for milk yield, fat-corrected milk yield, and milk contents. The milk and fat-corrected milk yield and somatic cell content (SCC) decreased from day 180 to 360, whereas the other milk contents increased from day 180 to 270 and then decreased from day 270 to 360. The effects of maternal treatment were detected (*p* ≤ 0.03) for plasma concentrations of urea and IGF-1 (Table 4). The plasma urea was greater (*p* = 0.03) for LHA vs. HHA cows, whereas the plasma IGF-1 was greater (*p* < 0.01) for HHA vs. LHA cows.

### 3.3. Preweaning (Days 150 to 390)

The effects of maternal treatment × the day of study were detected (*p* = 0.03) for preweaning offspring BW (Table 5). The offspring BW on day 150 did not differ (*p* = 0.66) between treatments. The offspring BW on days 270 and 390 was greater (*p* ≤ 0.05) for HHA vs. LHA offspring. The offspring ADG from day 150 to 270 was greater (*p* = 0.03) for HHA vs. LHA offspring and did not differ (*p* = 0.75) between maternal treatments from day 270 to 390. The offspring ADG from day 150 to 390 tended to be greater (*p* = 0.10) for HHA vs. LHA offspring. The effects of maternal treatment were not detected (*p* ≥ 0.13) for LMA, BFT, marbling, and RFT (Table 5).

The effects of maternal treatment × the day of the study were detected (*p* = 0.01) for plasma concentrations of glucose, which was observed to decrease from day 270 to 390 for HHA offspring, while LHA offspring maintained glucose levels during preweaning. The effects of maternal treatment × the day of the study were not detected (*p* ≥ 0.27) for plasma concentrations of urea, albumin, creatinine, total proteins, cholesterol, triglycerides, AAT, GGT, insulin, and IGF-1, but the effects of maternal treatment were detected (*p* ≤ 0.01) for plasma concentrations of GGT and insulin and tended to differ (*p* ≤ 0.07) for the plasma concentrations of urea and total proteins (Table 6). Plasma concentrations of GGT and insulin were greater (*p* ≤ 0.01) for LHA vs. HHA offspring, whereas the plasma concentrations of urea and total proteins tended to be greater (*p* ≤ 0.07) for HHA vs. LHA offspring.

## 4. Discussion

### 4.1. Maternal Performance

During the pre-calving period, seasonal changes to the quantity and quality of the forage were observed. In the current study, average monthly precipitation during the dry (day 0 to 70) and rainy (day 71 to 150) seasons were 11 and 70 mm, respectively. Therefore, the different structure and chemical composition of the forage were expected. Despite these differences, we managed to maintain different HA during prepartum by controlling the stocking rates. After the calving season, all cow–calf pairs were combined and transferred to a single pasture and managed similarly, and because of that, the differences in HA disappeared. In addition, the different herbage allowances altered the forage CP content, suggesting that LHA cows consumed a lower protein supply during pregnancy. In the last trimester of gestation, a Nellore cow must consume 1.04 g/day of CP to meet its nutritional requirements, but this value is not achieved in low-quality pastures [20]. In this current study, it was estimated that HHA cows consumed 86% of their CP requirements, while LHA cows consumed 72% [17]. A low CP intake can reduce bacterial growth in the rumen and, consequently, the digestion of neutral detergent fiber (NDF) and organic matter (OM) from pasture [21]. Therefore, under tropical/subtropical conditions, such as those observed herein, cows not only have nutritional restrictions on protein but also likely energy.

The increased herbage mass for HHA vs. LHA cows provided a greater opportunity for selective grazing and the greater consumption of nutrients from forage [22]. The required herbage mass to ensure animal gains is 2000 kg DM/ha, which also may cause animal selection [23]. In this current study, the herbage mass for HHA and LHA cows was always greater than 3000 kg DM/ha, which meant that it was consistently higher than the recommended level. High animal performance and forage intake occurred when there were high herbage allowances, and the forage nutritive value was high [24]. Previous studies have shown that a higher forage allowance increases the number of meals and may also reduce the time per meal [25,26]. In a previous study on the same site as this current study, Rodrigues et al. [6] worked with a forage allowance > 5.2 kg DM/kg of BW and a herbage mass > 4700 kg DM/ha for pregnant Nellore cows and observed that although the pasture had a moderate nutritional value, it did not cause a severe nutritional restriction for non-supplemented pregnant cows. In this study, the HHA pastures always had a herbage mass above 4700 kg DM/ha, except on day 70, and the forage allowance was greater than 5.2 kg DM/kg BW, which suggests that HHA cows were not restricted in nutrients from the pasture, although the LHA cows probably had nutritional restriction since their pasture was always below 4700 kg DM/ha, except on day 0, and the forage allowance was less than 5.2 kg DM/kg BW. Furthermore, the estimated voluntary forage intake at the beginning of the experiment was the same for low and high allowances (5.65 kg DM/day). Close to the calving season, HHA cows had an estimated forage intake of 5.49 kg DM/day (a 3% decrease relative to day 0), whereas LHA cows had an estimated forage intake of 5.02 kg DM/day (a decrease of 11% relative to day 0). The cows assigned to LHA likely had limited forage DM intake due to lower forage mass.

The lower performance of cows observed in this study (Table 2) indicates that forage intake was limited by nutritional and non-nutritional factors [27]. The lower average daily gain of cows grazing on LHA pastures can be explained by the imposed pasture conditions since a smaller forage mass, limiting the biting rate, negatively affects animal performance [28]. When herbage allowance is not a limiting factor (≥5.2 kg DM/kg BW; [6]), the stocking rate has little effect on individual animal performance due to unlimited opportunity for forage selection [29]. As the stocking rate increases, animal ADG decreases due to the increased animal competition for forage intake and fewer opportunities for forage selection [30]. In this situation, as the supply of energy provided by the pasture does not meet the energy requirements of the fetus during the prepartum period and the mammary gland following calving, the mobilization of body reserves is triggered [31]. The lower BCS at the calving of LHA vs. HHA cows is probably associated with a greater mobilization of fat reserves, as indicated by LMA and BFT data collection herein. Although cows reached physiological maturity, skeletal muscle tissue in the carcass is mobilized to meet the demands of pregnancy and maintenance [32] and maintain circulating amino acids [33].

The BW and BCS loss after the calving of HHA cows can be explained due to the greater BCS of HHA vs. LHA cows at calving and, consequently, the higher energy requirement for maintenance [34,35]. This postpartum alteration corroborates data from Ayres et al. [13] and Bohnert et al. [35], in which a greater BCS at calving resulted in greater postpartum BCS loss. At weaning, HHA cows were, on average, 16 kg heavier than those on LHA, reinforcing the importance of prepartum nutrition for maintaining or improving postpartum BW [35].

At the beginning of the suckling phase, the cow prioritizes milk production [36]. Thus, when nutrient supply is restricted, the dietary intake of nutrients is not sufficient to maintain basal metabolism and milk production. According to Lazzarini et al. [37], for the correct use of NDF in low-quality pastures, a minimum of 7% of forage CP is required. In this study, the forage mass during the suckling phase (day 150 to 390) remained at around 7332 kg of DM/ha, and CP concentrations were 7.91 and 3.89% during the rainy and dry seasons, respectively, indicating regular nutritional value, despite being below the requirements of cow–calf pairs [31].

Maternal herbage allowance during pregnancy did not impact the daily milk production of cows. The increased prepartum intake of dietary nutrients was associated with a different response in milk production and composition in dairy cows [38,39,40]. However, similar responses were not observed in beef cows [41,42,43]. Cow milk production increased in some, but not all, studies following improved maternal prepartum nutrition [3,42]. However, the corrected daily milk production was lower for LHA vs. HHA cows, probably due to the lower amount of body fat reserves available for milk production. In addition, it appears that LHA cows produced milk with lower concentrations of fat and total solids than HHA cows. The composition of milk is closely related to forage intake since the degradation of fibrous carbohydrates from pastures increases the production of acetic acid, which is an important precursor of fat in milk, while non-fibrous carbohydrates increase the production of propionic acid, which is the main precursor of lactose in milk [16]. The rumen-degradable protein (RDP) is used by rumen bacteria to produce the microbial protein (PMIC), which, when combined with the rumen-non-degradable protein (RNDP), provides the amino acids that are absorbed in the intestine and used for milk protein synthesis. Hence, the identification of changes in the milk components may indicate nutrient deficiencies or the lack of synchrony in the ruminal degradability of energy and protein in the diet of lactating cows [16,44,45]. The characteristics of milk composition have been used as indicators of milk quality and can affect progeny performance and system efficiency [42,46].

The maternal herbage allowance impacts prepartum plasma concentrations of IGF-1 and the urea of cows. The plasma urea results at d 140 suggest that HHA cows had greater ruminal N capture [47] and, thus, microbial CP supply, as HHA cows had reduced plasma urea concentrations. In addition, energy and protein intake were positively correlated with plasma concentrations of glucose and IGF-1 [48,49]. However, the outcomes of energy and protein supplementation on prepartum concentrations of glucose and IGF-1 in cows were inconsistent. The in utero availability of glucose and amino acids modulates fetal growth [50]. The maternal circulating concentrations of glucose [51] may alter the glucose supply of the fetus, whereas circulating IGF-1 is synthesized and regulated by the placenta, maternal and fetal tissues [52]. For instance, glucose, IGF-1 plasma concentrations, or both are increased during the late gestation of cows under the maternal supplementation of energy and protein [53,54,55].

It is important to reaffirm that the present study was designed to evaluate cow/calf pair performances and physiological responses and was not adequately powered to evaluate binary outcomes associated with pregnancy attainment, which requires a greater number of animals and pasture replicates to reach >80% power. The different herbage allowances and resulting BCS during the prepartum period were not sufficient to impact the cow pregnancy percentage, in agreement with the previous literature [54,55]. Furthermore, Carvalho et al. [56], evaluating multiparous Nellore cows, observed that the lower BCS at calving and at the breeding season negatively influences the pregnancy rate in artificial insemination (AI) when compared to cows with a higher BCS; however, this is at a lower intensity than in primiparous and secundiparous cows, demonstrating that multiparous cows are more resistant to the effect of a lower BCS at calving and negative energy balance in the first 80 days postpartum.

### 4.2. Offspring Performance

The negative energy balance of gestating beef cows is one of the stressors that may harm the placental environment and calf growth [57] and impact the postnatal performance of the offspring [58]. In the present study, offspring BW at birth did not show a difference due to maternal nutrition. The maternal supplementation of protein and energy during gestation (first, second, or third trimester of gestation) did not affect or increase offspring birth BW by an average of 3.2 kg [3].

Despite the same BW at birth, calves from HHA cows were heavier on days 270 and 390 compared to calves born from LHA cows, which may be associated with greater prepartum BCS and postpartum milk quality [42,46]. Milk production and composition, and the amount and nutritional value of forage supplied before and after birth, are the main factors modulating calf growth before weaning [59]. These findings suggest that the higher offspring performance during preweaning is related to postnatal nutrition, resulting from dam milk, which had its composition altered due to prepartum nutrition.

A disturbance of glucose–insulin homeostasis during fetal development may alter nutrient availability during the early postnatal growth and development in cattle. These responses are expected when the total nutrient [60,61,62] or specific components of the diet, such as protein or energy, are limited [63]. In the current study, calves from LHA cows had greater average plasma insulin concentrations than calves from HHA cows. Furthermore, the plasma concentrations of glucose from days 270 to 390 decreased in HHA calves and remained constant for LHA calves. However, these responses are variable, and it is known that, for grazing animals, there may be a misinterpretation of blood results due to the stress factors involved in taking the animals to the stockyard. Furthermore, additional studies ought to consider the assessment of glucose tolerance and the insulin sensitivity of prenatally programmed calves to determine if a poor prenatal environment causes a decline with age [64].

## 5. Conclusions

In conclusion, reducing maternal herbage allowance decreased cows’ BW and BCS at calving and the fat-corrected milk yield compared to high maternal herbage allowance during the prepartum period. Prepartum maternal herbage allowances did not affect the birth body weight of the offspring, but a high maternal herbage allowance increased offspring body weight at weaning compared to the low maternal herbage allowance, likely due to greater in utero nutrition and cow fat-corrected milk yield. Moreover, different herbage allowances altered cow plasma concentrations of IGF-1 near calving and offspring plasma concentrations of insulin and glucose. Overall, a higher herbage allowance can be implemented from mid-gestation until calving to increase the cow prepartum performance, postpartum milk yield and composition, and positively modulate offspring preweaning growth.

## Figures and Tables

**Figure 1 animals-14-00163-f001:**
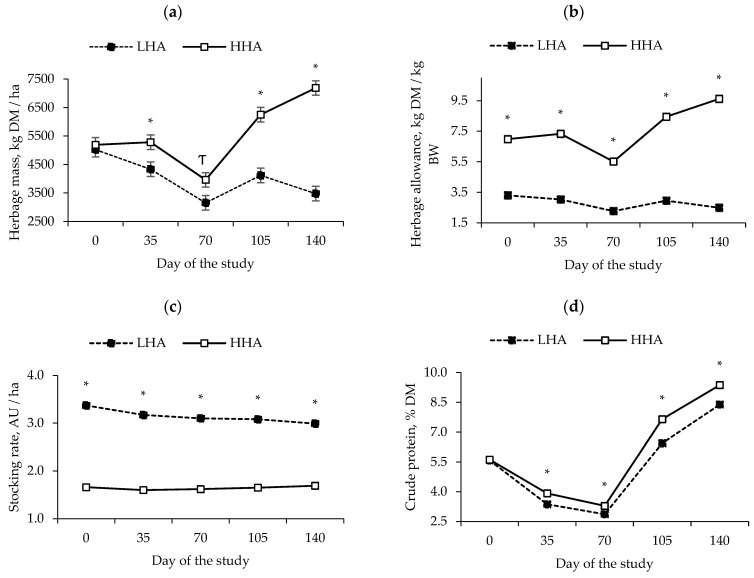
Herbage mass (**a**), herbage allowance (**b**), stocking rate (**c**), and crude protein (**d**) of pastures provided to beef cows from day 0 to 150 (5 cows and 7.5 ha/pasture). Treatments consisted of two herbage allowances during pregnancy: low (LHA; 2.80 kg DM/kg of BW) and high herbage allowance (HHA; 7.60 kg DM/kg of BW). Treatments were provided to cows from day 0 to 150 (150 ± 11 d prepartum until parturition). On day 150, all cow–calf pairs were combined in a single group, transferred to a single pasture (10 ha), and managed similarly with free choice and access to trace mineral salt supplementation until day 390. Values with asterisks (*) are significantly different (*p* ≤ 0.05), and Ƭ represents a tendency to be different (*p* = 0.08).

**Table 1 animals-14-00163-t001:** Average nutritional composition (DM basis) of supplements offered to cows from day 0 to 150 and trace mineral salt offered to cow–calf pairs from day 150 to 390.

Item	Protein Supplement Day 0 to 150	Trace Mineral Salt Day 150 to 390
Dry matter (DM), %	92	-
Crude protein, % of DM	50	-
Total digestible nutrients, % of DM	36	-
Mineral mixture *, % of DM	66 ^1^	100 ^2^
Target intake, % of BW	0.10	ad libitum

^1^ 50 g Ca (max); 35 g Ca (min); 20 g P (min); 80 g Na (min); 15 g S (min); 5000 mg Mg (min); 45 mg Co (min); 400 mg Cu (min); 25 mg I (min); 260 mg Mn (min); 9 mg Se (min); 1700 mg Z (min); 200 mg F (max); ^2^ 185 g Ca (max); 160 g Ca (min); 80 g P (min); 107 g Na (min); 12 g S (min); 5000 mg Mg (min); 107 mg Co (min); 1300 mg Cu (min); 70 mg I (min); 1000 mg Mn (min); 18 mg Se (min); 4000 mg Zn (min); 800 mg F (max). * Data provided by the manufacturer (Matsuda^®^, São Paulo, Brazil).

**Table 2 animals-14-00163-t002:** Body weight (BW), average daily gain (ADG), body condition score (BCS), ultrasound carcass measurements and reproductive performance of cows submitted to two different herbage allowances (low, LHA vs. high, HHA) from day 0 to 150 (150 ± 11 d prepartum until parturition; 6 pastures/maternal treatment; 3 to 5 cows and 7.5 ha/pasture) ^1^**.**

Item ^2^	Maternal Treatment		SEM	
	LHA	HHA		*p*-Value ^2^
Cow BW ^3^, kg				
Day 0	426	424	3.65	0.76
Day 130 (near calving)	407	442	3.65	<0.01
Day 390 (weaning)	399	415	3.65	<0.01
Cow ADG, kg/day				
Days 0 to 150	−0.155	0.141	0.05	<0.01
Days 150 to 390	−0.020	−0.100	0.03	0.01
Cow BCS ^3^				
Day 0	3.69	3.64	0.03	0.43
Day 35	3.51	3.39	0.03	0.07
Day 70	3.07	3.15	0.03	0.25
Day 130 (near calving)	3.06	3.28	0.03	<0.01
Day 150 (calving)	2.73	3.37	0.03	<0.01
Day 203 (start of breeding season)	2.89	2.98	0.03	0.19
Day 390 (weaning)	3.01	3.11	0.03	0.14
Cow LMA ^3^, cm^2^				
Day 0	57.6	57.7	0.86	0.94
Day 140 (near calving)	50.1	58.4	0.86	<0.01
Day 390 (weaning)	52.0	51.2	0.86	0.54
Cow BFT ^3^, mm				
Day 0	4.09	4.09	0.14	0.99
Day 140 (near calving)	3.33	4.18	0.14	<0.01
Day 390 (weaning)	3.24	3.18	0.14	0.80
Cow Marbling ^3^	3.80	3.92	0.08	0.15
Cow RFT ^3^, mm				
Day 0	6.82	6.88	0.30	0.90
Day 130 (near calving)	5.04	6.22	0.30	0.01
Day 390 (weaning)	4.47	4.53	0.30	0.90
Cow pregnancy rate, %	83.3	66.7	-	0.22

Abbreviations: BW = Body weight; ADG = Average daily gain; BCS = Body condition score; LMA = Longissimus muscle area; BFT = Backfat thickness; RFT = Rump fat thickness. ^1^ From day 150 to 390, all cow–calf pairs grazed on similar pastures and were provided with free choice access to trace mineral salt supplementation. ^2^
*p*-values for the comparison of treatments within each day for cow BW, BCS, LMA, BFT, and RFT, and *p*-values for the effects of treatment on cow ADG, marbling, and pregnancy rate. ^3^ Covariate-adjusted for respective data collected on day 0 (*p* < 0.01).

**Table 3 animals-14-00163-t003:** Milk production and composition of cows submitted to two different herbage allowances (low, LHA vs. high, HHA) from day 0 to 150 (150 ± 11 d prepartum until parturition; 6 pastures/maternal treatment; 3 to 5 cows and 7.5 ha/pasture) ^1^**.**

Item	Maternal Treatment		SEM	*p*-Value	Day of the Study			SEM	*p*-Value
	LHA	HHA		Maternal Treatment	180	270	360		Day of the Study
Milk yield, kg/day	4.36	4.77	0.37	0.28	5.85	4.82	3.02	0.24	<0.01
Fat-corrected milk yield, kg/day	4.08	4.90	0.38	0.03	5.61	5.01	2.84	0.36	<0.01
Milk fat, %	3.61	4.06	0.18	0.01	3.74	4.24	3.54	0.26	<0.01
Milk protein, %	3.44	3.59	0.10	0.14	3.25	3.82	3.48	0.06	<0.01
Milk lactose, %	4.63	4.61	0.12	0.89	4.70	4.76	4.40	0.08	0.02
Milk total solids, %	12.7	13.3	0.26	0.02	12.7	13.9	12.4	0.31	<0.01
Somatic cell contain, x mil/ml	272	241	115	0.79	420	189	115	76.3	<0.01
Milk urea N ^2^, mg/dL	10.9	10.9	0.43	0.86	10.8	11.5	10.4	0.30	0.01
Milk casein, %	2.70	2.85	0.09	0.10	2.51	3.09	2.72	0.05	<0.01

^1^ From day 150 to 390, all cow–calf pairs grazed on similar pastures and were provided with free choice access to trace mineral salt supplementation. ^2^ Covariate-adjusted for cow BW on day 0 (*p* = 0.02).

**Table 4 animals-14-00163-t004:** Plasma concentrations on day 140 of cows submitted to two different herbage allowances (low, LHA vs. high, HHA) from day 0 to 150 (150 ± 11 d prepartum until parturition; 6 pastures/maternal treatment; 3 to 5 cows and 7.5 ha/pasture).

Item	Maternal Treatment		SEM	*p*-Value
	LHA	HHA		Maternal Treatment
Glucose ^1^, mg/dL	82.3	82.6	4.22	0.94
Urea, mg/dL	33.1	27.8	2.32	0.03
Albumin, g/dL	3.93	3.88	0.23	0.80
Creatinine ^1^, mg/dL	2.04	1.86	0.13	0.16
Total protein, g/dL	9.25	8.25	0.68	0.15
Cholesterol, mg/dL	127	123	7.02	0.59
Triglycerides ^1^, mg/dL	36.9	33.8	3.04	0.31
Amino aspartate-transferase, U/L	89.2	83.9	6.40	0.41
Gamma-glutamyl transferase ^1^, U/L	17.9	18.8	1.22	0.50
Insulin, ulU/mL	13.8	10.8	2.03	0.15
IGF-1 ^1^, ng/mL	194	237	13.6	<0.01

^1^ Covariate-adjusted for the respective plasma data obtained on day 0 (*p* < 0.05).

**Table 5 animals-14-00163-t005:** Preweaning body weight (BW) adjusted for 205 days of age with average daily gain (ADG) and carcass ultrasound measurements of calves born to cows submitted to two different herbage allowances (low, LHA vs. high, HHA) from day 0 to 150 (150 ± 11 d prepartum until parturition; 6 pastures/maternal treatment; 3 to 5 cows and 7.5 ha/pasture) ^1^.

Item	Maternal Treatment		SEM	
	LHA	HHA		*p*-Value ^2^
Adjusted body weight, kg				
Birth	32.5	33.1	2.30	0.66
Day 270	122	131	2.30	0.03
Day 390 (weaning)	169	178	2.30	0.05
Average daily gain, kg/day				
Days 150 to 270	0.74	0.82	0.03	0.03
Days 270 to 390	0.51	0.51	0.02	0.75
Days 0 to 390	0.66	0.70	0.02	0.10
*Longissimus* muscle area, cm^2^	35.0	37.1	1.36	0.13
Backfat thickness, mm	2.34	2.35	0.13	0.95
Marbling	2.63	2.98	0.21	0.10
Rump fat thickness, mm	3.59	3.84	0.17	0.14

^1^ From day 150 to 390, all cow–calf pairs grazed on similar pastures and were provided with free choice access to trace mineral salt supplementation. ^2^
*p*-value for the comparison of treatments within the day for calf-adjusted BW, and *p*-value for the effects of treatment for all remaining variables.

**Table 6 animals-14-00163-t006:** Plasma concentrations on days 270 and 390 for calves born to cows submitted to two different herbage allowances (low, LHA vs. high, HHA) from day 0 to 150 (150 ± 11 d prepartum until parturition; 6 pastures/maternal treatment; 3 to 5 cows and 7.5 ha/pasture) ^1^.

Item	Maternal Treatment		SEM	
	LHA	HHA		*p*-Value ^2^
Glucose, mg/dL				
Day 270	89.9	94.2	3.91	0.29
Day 390 (weaning)	87.8	79.8	3.91	0.12
Urea, mg/dL	10.8	12.4	0.88	0.07
Albumin, g/dL	4.15	4.17	0.10	0.82
Creatinine, mg/dL	1.79	1.75	0.07	0.54
Total proteins, g/dL	7.75	8.24	0.26	0.06
Cholesterol, mg/dL	161	161	7.58	0.98
Triglycerides, mg/dL	39.1	36.7	2.90	0.39
Amino aspartate-transferase ^3^, U/L	88.3	86.2	3.92	0.60
Gamma-glutamyl transferase, U/L	22.0	19.1	1.14	0.01
Insulin, ulU/mL	7.49	4.97	0.89	<0.01
IGF-1, ng/mL	172	186	16.5	0.39

^1^ From day 150 to 390, all cow–calf pairs grazed on similar pastures and were provided with free choice access to trace mineral salt supplementation. ^2^
*p*-value for the comparison of treatments within the day for plasma glucose, and *p*-value for the effects of treatment for all remaining variables. ^3^ Covariate-adjusted for cow BW on day 0 (*p* < 0.01).

## Data Availability

Data are available on request due to privacy or ethical restrictions.

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
