# Peer review of "Effect of Different Herbage Allowances from Mid to Late Gestation on Nellore Cow Performance and Female Offspring Growth until Weaning"

_animals, 2024, doi:10.3390/ani14010163_

Round 1
Reviewer 1 Report
Comments and Suggestions for Authors
“Effect of Different Herbage Allowances from Mid to Late Gestation on Nellore Cows Performance and the Female Offspring Growth until Weaning”.
The authors aimed to evaluate the effects of contrasting HA (low vs. high) from mid-gestation to calving on performance, physiology and milk production of Nellore cows and postnatal growth and physiology of their offspring.
Below are presented the observations to improve the quality of the paper:
The introduction should be broadened and deepened. I suggest some articles to read and integrate into your introduction.
1. Effect of grazing system on fetal development in Nellore cattle https://doi.org/10.1016/j.theriogenology.2003.07.021
2. Effects of pre- and postpartum supplementation on lactational and reproductive performance of grazing Nellore beef cows
https://doi.org/10.1071/AN18251
3. Effect of supplementation level on performance of growing Nellore and its influence on pasture characteristics in different seasons
https://doi.org/10.1080/1828051X.2018.1504633
In general the part of the experimental protocols must be rewritten and made clearer.
L76-88: please write the experimental plan again: the groups and diets and the organization of the pastures are not well understood.
It's not clear: was the herb available only Brachiaria brizantha? otherwise if the animals also had other herbaceous species available, they must be described as they are and a characterization of the pasture given.
L89: Review how you wrote the time.
L108, Forage and feed: How did you take pasture samples? describe the criteria and guidelines you followed; example: what tools did you use for the sampling? How much did the sample taken weigh? How many portions of pasture did you take? Etc
L117: "..individual shrunk BW of cows were assessed..."how was it assessed? with scale?
L121: only 3 cows for blood tests seems like a low number to me.
L179 review the formatting of the method you cite
L180 controls spaces between words (double spaces).
L 220: in the results when you recall the figures, also insert the letter (to help the reader) and not just the number of the figure. for example: figure 1 a.
L244 replace (p ≥ 0.14) with the value of p, example: p=0.14
L289 Can you say something about colostrum? about quality? The future of the calves in terms of weight and growth may also depend on this
L443 Be careful, it's probably also the fault of the hiring team. did you record this data? furthermore it could also be influenced by the genetics of the fathers (who were different). Please clarify this.
L450 I believe that this part needs to be explored further: if you decide to include table 6 with the results, it is assumed that constipation and therefore the sample size are considered valid. Therefore, parameters that are statistically significant must be discussed and provided with a possible explanation, as for example in the case of insulin.
L463 In the conclusions I would be more cautious when you talk about herbs and their effects because you did not take these herbs into consideration nor did you mention what they were. you could add this part in the materials and methods citing some of these herbs and their effects in the literature.
Author Response
- Summary
Thank you very much for taking the time to review this manuscript. Please find the detailed responses below and the corresponding corrections highlighted (in red) in the re-submitted file.
- Point-by-point response to Comments and Suggestions for Authors
Comments 1: The introduction should be broadened and deepened. I suggest some articles to read and integrate into your introduction.
Response 1: Thank you for pointing this out. I agree with this comment. Therefore, I have modified some points (in red) and references.
Comments 2: L76-88: please write the experimental plan again: the groups and diets and the organization of the pastures are not well understood.
Response 2: Agree. I have modified to emphasize this point (in red).
Comments 3: It's not clear: was the herb available only Brachiaria brizantha? otherwise if the animals also had other herbaceous species available, they must be described as they are and a characterization of the pasture given.
Response 3: Yes, it was only Brachiaria brizantha; but the correct name is: Urochloa brizantha. I have corrected in the text.
Comments 4: L89: Review how you wrote the time.
Response 4: Thank you.
Comments 5: L108, Forage and feed: How did you take pasture samples? describe the criteria and guidelines you followed; example: what tools did you use for the sampling? How much did the sample taken weigh? How many portions of pasture did you take? Etc
Response 5: Agree. I have modified to emphasize this point (in red).
Comments 6: L117: "..individual shrunk BW of cows were assessed..."how was it assessed? with scale?
Response 6: Thank you. I have modified to emphasize this point (in red).
Comments 7: L121: only 3 cows for blood tests seems like a low number to me.
Response 7: It was 18 cows per treatment, totalizing 36 cows. I have modified to emphasize this point (in red).
Comments 8: L179 review the formatting of the method you cite
Response 8: Thank you.
Comments 9: L180 controls spaces between words (double spaces).
Response 9: Thank you.
Comments 10: L 220: in the results when you recall the figures, also insert the letter (to help the reader) and not just the number of the figure. for example: figure 1 a.
Response 10: Thank you. I have modified to emphasize this point (in red).
Comments 11: L244 replace (p ≥ 0.14) with the value of p, example: p=0.14
Response 11: Thank you. However, in this sentence: “Cow BCS on days 0, 70, 203 and 390 did not differ (p ≥ 0.14)” I refer to the BCS of several days, and not just of a specific day. So, the BCS did not differ, because p ≥ 0.05, and the lowest P-value on those days mentioned was 0.14, therefore p ≥ 0.14.
Comments 12: L289 Can you say something about colostrum? about quality? The future of the calves in terms of weight and growth may also depend on this.
Response 12: Unfortunately, in this study we did not measure colostrum quality. As they were grazing Nellore cows, for colostrum to be collected, we would need to take these animals to the stockyard immediately after calving. Furthermore, the colostrum is the first suckling, so we would have to ensure that the calf did not suckle the colostrum, and this could harm the calf's colostrum intake. Another way of measuring the quality of colostrum would be through blood collection from calves and analysis of antibodies (immunity). However, collecting the colostrum samples would only be possible in the stockyard, and due to the stress factors involved in taking these animals to the stockyard, there may be confusion in immune responses. That is why this assessment was not carried out.
Comments 13: L443 Be careful, it's probably also the fault of the hiring team. did you record this data? furthermore it could also be influenced by the genetics of the fathers (who were different). Please clarify this.
Response 13: Yes, we registered. Two AIs were performed, and we found no statistical difference. I have modified this point (in red).
Comments 14: L450 I believe that this part needs to be explored further: if you decide to include table 6 with the results, it is assumed that constipation and therefore the sample size are considered valid. Therefore, parameters that are statistically significant must be discussed and provided with a possible explanation, as for example in the case of insulin.
Response 14: Thank you for pointing this out. I agree with this comment. I have modified this point (in red).
Comments 15: L463 In the conclusions I would be more cautious when you talk about herbs and their effects because you did not take these herbs into consideration nor did you mention what they were. you could add this part in the materials and methods citing some of these herbs and their effects in the literature.
Response 15: Thank you. But, I mentioned in the material and methods that to maintain high and low herbage allowance, I adjusted the stocking rate with replacement animals. Our objective was to evaluate the effects of contrasting herbage allowances (low vs. high) from mid-gestation in Nellore cows. So, in the conclusions I referred to the herbage allowances that I tested in my research.
Reviewer 2 Report
Comments and Suggestions for Authors
Nice work conducting the experiment and writing the manuscript.
1. What binary data did you work with?
2. Since table 3 is too long, you may want to change the layout of that page to horizontal to maintain the margins
Comments on the Quality of English LanguageEnglish seems to be okay
Author Response
- Summary
Thank you very much for taking the time to review this manuscript. Please find the detailed responses below and the corresponding revisions highlighted (in green) in the re-submitted file.
- Point-by-point response to Comments and Suggestions for Authors
Comments 1: What binary data did you work with?
Response 1: Thank you for pointing this out. I have modified to emphasize this point (in green). It was the reproductive parameters (cow pregnancy rate).
Comments 2: Since table 3 is too long, you may want to change the layout of that page to horizontal to maintain the margins
Response 2: Agree. I changed in the file.
Reviewer 3 Report
Comments and Suggestions for Authors
This manuscript reveals that reducing maternal herbage allowance decreases cow prepartum performance, post-partum milk yield and milk composition and increases female offspring preweaning growth. The results are meaningful. However, the methods have some problems.
I only have the following comments:
Materials and methods:
L100 Table 1: Is the ratio of dietary formulations calculated or measured in the laboratory?
L103 Please correct “1700mg Z (min)”.
L279-280 Table 3: Which group are “Day of 180 270 360” belong to? LHA or HHA?
References:
Delete all links of all references and standardize the format of references.
Author Response
- Summary
Thank you very much for taking the time to review this manuscript. Please find the detailed responses below and the corresponding corrections highlighted (in blue) in the re-submitted file.
- Point-by-point response to Comments and Suggestions for Authors
Comments 1: L100 Table 1: Is the ratio of dietary formulations calculated or measured in the laboratory?
Response 1: The ratio of dietary was provided by the manufacturer (Matsuda®), it was done in its laboratory.
Comments 2: L103 Please correct “1700mg Z (min)”.
Response 2: Thank you.
Comments 3: L279-280 Table 3: Which group are “Day of 180 270 360” belong to? LHA or HHA?
Response 3: Days 180, 270 and 360 belong to both, LHA and HHA, as it is the period effect, without treatment effect. Corresponds to the average for the period.
Comments 4: Delete all links of all references and standardize the format of references.
Response 4: Thank you for pointing this out.
Round 2
Reviewer 1 Report
Comments and Suggestions for Authors
The authors responded precisely to all my comments, making the requested corrections. For this reason this manuscript can be accepted. Thank you